# EBP50 Depletion and Nuclear β-Catenin Accumulation Engender Aggressive Behavior of Colorectal Carcinoma through Induction of Tumor Budding

**DOI:** 10.3390/cancers16010183

**Published:** 2023-12-29

**Authors:** Takashi Itou, Yu Ishibashi, Yasuko Oguri, Miki Hashimura, Ako Yokoi, Yohei Harada, Naomi Fukagawa, Misato Hayashi, Mototsugu Ono, Chika Kusano, Makoto Saegusa

**Affiliations:** 1Department of Pathology, Kitasato University School of Medicine, 1-15-1 Kitasato, Minami-ku, Sagamihara 252-0374, Kanagawa, Japan; dm21003@st.kitasato-u.ac.jp (T.I.); y-ishiba@kitasato-u.ac.jp (Y.I.); oguriy@med.kitasato-u.ac.jp (Y.O.); mhashimu@med.kitasato-u.ac.jp (M.H.); ayokoi@med.kitasato-u.ac.jp (A.Y.); harada.yohei@st.kitasato-u.ac.jp (Y.H.); fukagawa.naomi@st.kitasato-u.ac.jp (N.F.); hayashi.misato@st.kitasato-u.ac.jp (M.H.); m.ono@insti.kitasato-u.ac.jp (M.O.); 2Department of Gastroenterology, Kitasato University School of Medicine, 1-15-1 Kitasato, Minami-ku, Sagamihara 252-0374, Kanagawa, Japan; c-kusano@kitasato-u.ac.jp

**Keywords:** EBP50, β-catenin, colorectal carcinoma, tumor budding

## Abstract

**Simple Summary:**

Here, we examined the functional role of Ezin-radixin-moesin-binding phosphoprotein 50 (EBP50) during colorectal carcinoma (CRC) progression. EBP50 depletion was frequently found in the outer areas of tumor lesions and was significantly associated with several unfavorable prognostic factors including depressive tumor growth, deep invasions, high tumor budding (BD), and the presence of neural invasion in CRC cases. EBP50-knockout CRC cells disrupted the interaction between EBP50 and β-catenin at the plasma membrane, leading to epithelial–mesenchymal transition (EMT)-like features, decreased proliferation, accelerated migration capability, and stabilized nuclear β-catenin. In addition, Slug expression was positively correlated with nuclear β-catenin status in tumor outer lesions including BD areas, consistent with β-catenin-driven transactivation of the Slug promoter. These findings suggest that EBP50 depletion is due to nuclear β-catenin accumulation, allowing transactivation of the Slug gene to promote EMT. This in turn contributes to the progression of CRC to a more aggressive phase through the induction of tumor budding.

**Abstract:**

Ezin-radixin-moesin-binding phosphoprotein 50 (EBP50) is a scaffold protein that interacts with several partner molecules including β-catenin. Here, we examined the crosstalk between EBP50 and nuclear catenin during colorectal carcinoma (CRC) progression. In clinical samples, there were no correlations between the subcellular location of EBP50 and any clinicopathological factors. However, EBP50 expression was significantly lower specifically in the outer areas of tumor lesions, in regions where tumor budding (BD) was observed. Low EBP50 expression was also significantly associated with several unfavorable prognostic factors, suggesting that EBP50 depletion rather than its overexpression or subcellular distribution plays an important role in CRC progression. In CRC cell lines, knockout of EBP50 induced epithelial–mesenchymal transition (EMT)-like features, decreased proliferation, accelerated migration capability, and stabilized nuclear β-catenin due to disruption of the interaction between EBP50 and β-catenin at the plasma membrane. In addition, Slug expression was significantly higher in outer lesions, particularly in BD areas, and was positively correlated with nuclear β-catenin status, consistent with β-catenin-driven transactivation of the Slug promoter. Together, our data suggest that EBP50 depletion releases β-catenin from the plasma membrane in outer tumor lesions, allowing β-catenin to accumulate and translocate to the nucleus, where it transactivates the Slug gene to promote EMT. This in turn triggers tumor budding and contributes to the progression of CRC to a more aggressive phase.

## 1. Introduction

Colorectal carcinoma (CRC) is the third most diagnosed cancer after lung and breast carcinomas [1]. There has been a rapid increase in its incidence over recent decades in Asia-Pacific populations; this may be due to attribution to environmental factors such as aging and the adoption of the Western diet [2]. In fact, the 5-year prevalence and age-standardized incidences in certain Asian countries such as Japan and Korea are higher than those reported in Western countries [3].

Ezrin-radixin-moesin (ERM)-binding phosphoprotein 50 (EBP50) is a 55-kDa scaffold phosphoprotein and is highly expressed in several epithelial tissues and localized at the apical plasma membrane of the polar epithelia, where it regulates apical microvilli formation [4,5]. Apico-basal cell polarity is a morphological characteristic disrupted early in the development of epithelial malignancies [6]. Interestingly, EBP50 adopts a tumor suppressor role when present in the apical plasma membrane, but is oncogenic once it relocalizes from the membrane to the cytoplasm and nucleus [7].

Disorganization of epithelial morphology triggers the epithelial–mesenchymal transition (EMT), which in turn increases tumor cell invasiveness [8]. The EMT is characterized by various degrees of epithelial changes from mild loss of cell polarity to the acquisition of frank mesenchymal/fibroblastic characteristics with increased cell motility [9]. In CRC, tumor budding (BD) is the first step in EMT, lymphovascular invasion, lymph node metastasis, and spreading to distant organs [10]. Although EBP50 has a functional role in EMT [11], little is known about its association with BD features in CRC.

EBP50 can form protein complexes at adhesion junctions by directly interacting with β-catenin [12,13]. EBP50 also promotes the β-catenin/TCF-1B interaction, thereby forming a ternary complex that is recruited to the Wnt-responsive gene [14]. Given that approximately 85% of sporadic CRCs have deregulated canonical Wnt/β-catenin signaling [15], we hypothesized there would be a functional interplay between EBP50 and β-catenin during CRC progression through modulation of EMT-related BD features. To test this, we investigated the expression and localization of EBP50 and β-catenin in CRC cell lines and clinical samples.

## 2. Materials and Methods

### 2.1. Clinical Cases and Evaluation of Tumor Budding Findings

We retrospectively evaluated pathological specimens from 100 consecutive CRC patients who had undergone surgery between 2018 and 2022, at Kitasato University Hospital, according to the criteria of the Japanese Classification of Colorectal, Appendiceal, and Anal Carcinoma (JCCAAC) and the TNM classification [16,17]. Twenty cases of normal colorectal mucosa, low-grade tubular adenomas, high-grade tubular adenoma, and carcinoma in adenoma, respectively, were also investigated. All tissues were routinely fixed in 10% formalin and processed for embedding in paraffin wax. This study was approved by the Kitasato University Medical Ethics Committee (B21-214).

Tumor budding (BD) was defined as the presence of individual and/or small groups of tumor cells at the invasive fronts as described previously [18]. Briefly, the BD score was evaluated according to the criteria of the JCCAAC [16]. The BD features were then subcategorized into three groups as follows: BD:1, less than four individual and/or small groups of tumor cells at the invasive fronts; BD:2, five to nine; and BD:3, more than ten.

### 2.2. Antibodies

Anti-phospho (p) Rb at Ser807/811 (catalog number #9308) and anti-Slug antibodies (#9585) were purchased from Cell Signaling Technology (Danvers, MA, USA). Anti-EBP50 (ab109430), Anti-cyclin A2 (#ab181591), anti-Snail (#ab180714), and anti-Twist1 (#ab50887) antibodies were obtained from Abcam (Cambridge, MA, USA). Anti-E-cadherin, anti-Ki-67 (#M7240), anti-p21^waf1^ (#M7202), and anti-cyclin D1 antibodies (#M3635) were from Dako (Copenhagen, Denmark). Anti-p27^kip1^ (#610242), anti-Rb (#554136), anti-aldehyde dehydrogenase 1 (ALDH1) (#611194), anti-β-catenin (#610154), and anti-N-cadherin antibodies (#610920) were from BD Biosciences (San Jose, CA, USA). Anti-FLAG M2 (#F3165), anti-ZEB1 (#HPA027524), and anti-β-actin antibodies (#HPA025958) were from Sigma-Aldrich Chemicals (St. Louis, MO, USA). Anti-vimentin (sc-6260) and anti-cyclin B1 antibodies (#sc-752) were from Santa Cruz Biotech (Santa Cruz, CA, USA). Anti-E-cadherin antibody (#M106, Takara Bio Inc., Siga, Japan) was purchased from Cell Signaling Technology (Danvers, MA, USA).

### 2.3. Immunohistochemistry (IHC) Quantification

IHC, calculation of IHC scores, and region-specific distribution (Appendix A) for the proteins of interest were performed as described previously [19,20]. Outer lesions were further subdivided into two categories including BD and non-BD areas on the basis of tumor budding features. For EBP50, cases were defined as membrane (Me)- or cytoplasm (Cyt)-immunopositive when either Me or Cyt staining was observed in more than 50% of cells. In outer lesions, reduced EBP50 immunoreactivity was subdivided into four categories as follows: none, the depletion of immunoreactivity is found in less than 1% of cells; mild, 1–19%; moderate, 20–39%; severe, more than 40%. High and low EBP50 depletion were derived by aggregating scores from severe/moderate and mild/none groups, respectively.

### 2.4. Immunofluorescence

Slides of CRC tissues were heated in 10 mM citrate buffer (pH 6.0) for 3 × 5-min cycles using a microwave oven and then incubated overnight with the primary antibodies. EBP50-knockout (EBP50-KO) and mock cells were also incubated with anti-β-catenin and anti-EBP50 antibodies. Alexa 488 and 570 (Thermo Fisher Scientific, Waltham, MA, USA) were used as secondary antibodies as described previously [21,22].

### 2.5. Plasmids and Cell Lines

Glutathione S-transferase (GST)-fusion protein constructs including full length, PDZ1, PDZ2, and EB domains, pcDNA3.1-β-catenin delS45, p3xFLAG-CMV14-EBP50, and -2125/-235 pGL3B-Slug were used as described previously [21,22,23].

CRC cell lines, HCT116, OUMS23, and DLD-1, were obtained from the American Type Culture Collection (Manassas, VA, USA) and the JCRB Cell Bank (National Institute of Biomedical Innovation, Osaka, Japan), respectively. The EBP50-KO cell line was generated using HCT116 cells (Appendix A). The guide RNA sequence (gRNA: 5′-GAGAAGGGTCCGAACGGCTACGG-3′) was used. The complementary oligonucleotides for gRNA were annealed and cloned into pSpCas9n(BB)-2A-Puro (PX459) V2.0 (#62988) (Addgene, Watertown, MA, USA). The pSpCas9n(BB)-2A-Puro (PX459) V2.0/gRNA construct was transfected into HCT116 cells and EBP50-KO lines were established as described previously [22].

### 2.6. GST Pull-Down Assay

GST-EBP50-full length, GST-EBP50-PDZ1, GST-EBP50-PDZ2, and GST-EBP50-EB were induced by 1 mM isopropyl-β-D-thiogalactopyranoside and purified with glutathione-sepharose beads. HCT116 cell lysates were mixed with purified GST-EBP50-full length, GST-EBP50-PDZ1, GST-EBP50-PDZ2, or GST-EBP50-EB immobilized on the beads. Pull-down assays were performed as described previously [21].

### 2.7. Transfection

Transfection was carried out using LipofectAMINE PLUS (Invitrogen, Carlsbad, CA, USA). Luciferase activity was assayed as described previously [21,22,23].

### 2.8. Western Blot Assays and Co-Immunoprecipitation Assays

Total cellular proteins were isolated using RIPA buffer (20 mM Tris-HCl (pH 7.2), 1% Nonidet P-40, 0.5% sodium deoxycholate, 0.1% sodium dodecyl sulfate). The cytoplasmic, membranous, nuclear, and cytoskeletal fractions were prepared using ProteoExtract Subcellular Proteome Extraction kit (Merk KGaA, Darmstadt, Germany). Western blot assay was performed as described previously [21,22,23]. To examine the ratios of β-catenin relative to β-actin, the signals were analyzed using ImageJ software version 1.41 (NIH, Bethesda, MD, USA; http://imageJ.nih.gov/ij, last accessed on 4 May 2021). For co-immunoprecipitation, FLAG-EBP50 was transfected into HCT116 cells. After subdivision of transfected cells into three tubes, cells were lysed with IP buffer (10 mM Tris-HCl (pH 7.6), 100 mM NaCl, 1% NP-40). Cell lysates were cleared and incubated with mouse IgG, anti-β-catenin, or anti-FLAG antibodies, followed by incubation with Protein G-Sepharose (Amersham Pharmacia Biotechnology, Piscataway, NJ, USA) as described previously [21,22,23].

### 2.9. Flow Cytometry and Aldefluor Assay

Cells were fixed using 70% alcohol and stained with propidium iodide (Sigma, St. Louis, MO, USA) for cell cycle analysis. ALDH1 enzyme activity in viable cells was determined using a fluorogenic dye-based Aldefluor assay (Stem Cell Technologies, Grenoble, France) according to the manufacturer’s instructions. The prepared cells were analyzed by flow cytometry using BD FACS Calibur (BD Biosciences, Franklin Lakes, NJ, USA) and CellQuest Pro software version 3.3 (BD Biosciences) as described previously [21,22].

### 2.10. Wound Healing Assay

Cells were seeded into 24-well tissue culture plates and grown to reach almost total confluence. After a cell monolayer formed, a wound was scratched with a sterile 200 μL tip. The area of the wound was also analyzed using ImageJ software version 1.41 (NIH). Cell migration parameters were calculated in pixels as wound closure as described previously [21,22].

### 2.11. Migration Assay

Cell migration was determined using 24-well Transwell chambers with 8 μm pore size (Corning, New York, NY, USA). The lower chamber was filled with medium containing 10% serum. Cells were suspended in serum-free upper medium and planed into the upper chamber. After 24 h, the number of cells stained by hematoxylin-eosin (HE) on the bottom surface of the polycarbonate membranes was counted using a light microscope as described previously [21,22].

### 2.12. Spheroid Assay

Cells (×10^3^) were plated in low cell binding plates (Thermo Fisher Scientific, Yokohama, Japan) in Cancer Stem Cell Premium (ProMab Biotech, Richmond, CA, USA). Uniform spheroids of at least 50 μm in diameter were counted approximately two weeks after plating as described previously [21,22].

### 2.13. TCGA Data Analysis

The Cancer Genome Atlas (TCGA) UCS annotated EBP50 mRNA expression data (RNA Seq V2 PSEM) was extracted from cBioportal for Cancer Genomics (http://www.cbioportal.org/, last accessed on 9 August 2023) for 588 CRC cases. The data were subcategorized into high and low groups based on the Z score (cut-off value was 1.0 and 1.5) for mRNA expression levels.

### 2.14. Statistical Analysis

Comparative data were analyzed using the Mann–Whitney U-test, Chi-square test, or Spearman’s rank correlation coefficient, as appropriate. Overall survival (OS) and progression-free survival (PFS) were calculated as described previously [21,22]. OS and PFS were estimated using the Kaplan–Meier method, and statistical comparisons were made using the log-rank test. The cut-off for statistical significance was set as *p* < 0.05.

## 3. Results

### 3.1. EBP50 Expression during Adenoma-Carcinoma Progression and TCGA Data Analysis in CRC

EBP50 immunoreactivity was observed in the apical plasma membrane (Me-EBP50) and/or cytoplasmic compartment (Cyt-EBP50) of epithelial cells during adenoma to carcinoma progression in CRC (Appendix A). Me-EBP50 scores did not correlate with the grade of adenoma. In contrast, there was a stepwise increase in Cyt-EBP50 scores from low- through high-grade adenomas to carcinoma in adenoma (non-invasive CRC) lesions. Cyt-EBP50 score was also significantly higher in normal as compared to low-grade adenoma lesions (Appendix A). Although there was a heterogeneous distribution of Me- and Cyt-EBP50 positivity within CRC tissues (Appendix A), Me- and Cyt-EBP50 immunopositivity predominated in 41% and 59% of the 100 CRC cases, respectively. With the exception of tumor invasion into lymph vessels, neither membrane nor cytoplasmic localization of EBP50 correlated with the clinicopathological samples we examined (Table 1). In addition, we used TCGA data to evaluate the prognostic significance of EBP50 expression in 588 CRC cases. Following the separation of cases into EBP50-high and -low groups, Kaplan–Meier curves analysis showed that EBP50 expression had no prognostic significance for OS or PFS (Appendix A).

### 3.2. Reduced EBP50 Expression in Outer Lesions of CRC Is Associated with Aggressive Tumor Behavior

Decreased EBP50 immunostaining was frequently observed in BD areas of outer lesions (Figure 1A), where it was accompanied by strong nuclear β-catenin and a lack of Ki-67 immunostaining. This observation was reflected by IHC scores (Figure 1B,C), confirming that the EBP score was positively and negatively correlated with Ki-67 and nuclear β-catenin scores, respectively (Table 2).

We next examined the correlations between EBP50, β-catenin, and Ki-67 staining and determined whether there was region-specific expression of these markers. EBP50 (including Me- and/or Cyt-EBP50 subtypes) and Ki-67 scores were significantly lower in the outer than the inner and middle areas of lesions, in contrast to the significantly higher nuclear β-catenin scores in the outer area (Appendix A, Figure 2). In the outer lesions, 64% of cases had high (severe/moderate) EBP50 depletion and 36% had low (mild/none) EBP50 depletion. Compared to the low EBP50 depletion group, the high EBP50 depletion group was significantly associated with depressive tumor growth, deep invasions, high BD scores, and the presence of neural invasions. The level of EBP50 expression between the Me- and Cyt-EBP50 categories was similar (Table 1).

### 3.3. Knockout of EBP50 Is Associated with Decreased Proliferation and Accelerated Migration Capability

To examine the effect of EBP50 depletion in CRC cells, we first established an EBP50-KO HCT116 cell line clone, EBP50-KO#30 (Appendix A). EBP50-KO#30 cells demonstrated a significant switch towards an EMT-like fibroblastic morphology (Figure 3A). The KO cells tended to proliferate more slowly than mock cells, particularly in the exponential growth phase; there were also proportionally fewer cells in the S phase and more cells in the G2/M phase (Figure 3B). We next examined the expression of several cell cycle-related molecules during cell growth, by rendering EBP50-KO#30 cells quiescent by serum starvation before stimulating with serum. At 9 and 24 h after release into the cell cycle, lower levels of Rb and cyclin A2 and higher p27^kip1^ levels were observed in EBP50-KO#30 cells relative to the mock cells, whereas expression of pRb, cyclin B1, and cyclin D1 was unchanged (Figure 3C).

The EBP50-KO#30 cells also had significantly increased migration rates as compared to the mock cells (Figure 4A) and refilled wounded empty spaces more rapidly (Figure 4B), consistent with their EMT-like features. This was accompanied by increased expression of the EMT-related molecule, Slug, but not Snail, ZEB1, or Twist1; there were no differences in the expression of E-cadherin and Vimentin (Figure 4C). ALDH1 expression was much lower in EBP50-KO#30 cells than mock cells, and this was consistent with reduced CSC-like ALDH^high^ population upon EBP50 knockout (Figure 4C,D). In addition, there was a significant decrease in the number of well-defined, round spheroids > 50 μm in diameter in EBP50-KO cells (Figure 4E).

These findings suggest that knockout of EBP50 induces an EMT-like phenotype, decreases proliferation, and accelerates migration in CRC cells, but does not induce CSC-like properties.

### 3.4. EBP50 Strongly Interacts with β-Catenin at the Plasma Membrane

We observed that β-catenin could be co-immunoprecipitated with EBP50 but not vice versa (Figure 5A). To map the β-catenin binding region within EBP50, we performed GST pull-down assays with full-length and truncated forms of EBP50 (Figure 5B). This revealed that β-catenin specifically bound to PDZ2 (but not the PDZ1 or EB domains) (Figure 5C).

To examine the subcellular localization of EBP50 and β-catenin, we performed immunofluorescence analysis using EBP50-KO#30 and mock cells. Nuclear β-catenin accumulation was frequently observed in EBP50-KO#30 cells, whereas EBP50 and β-catenin colocalized predominantly at the plasma membrane in mock cells (Figure 5D). The number of nuclear β-catenin-positive cells was significantly higher in EBP50-KO#30 cells compared to mock cells (Figure 5E), which was consistent with the threefold increase in the nuclear β-catenin fraction of KO cells, as well as membranous and cytoskeletal fractions (Figure 5F).

Because we observed Slug expression in EBP50-KO#30 cells, we next examined whether nuclear β-catenin was associated with Slug expression in CRC tissues. Nuclear β-catenin and Slug frequently colocalized in outer lesions, particularly in BD areas (Figure 6A). An examination of regional expression revealed that the Slug score was significantly higher in the outer area of tumors (Figure 6B) and was positively correlated with the nuclear β-catenin score (Table 1). The *Slug* promoter was activated about 2- to 3-fold upon transfection of β-catenin, and this effect was further enhanced by cotransfection of p300 (Figure 6C).

Together, these findings suggest that EBP50 can interact with β-catenin at the plasma membrane, and that EBP50 depletion engenders stabilization of nuclear β-catenin. Ultimately this leads to β-catenin-dependent transactivation of the *Slug* promoter.

## 4. Discussion

Here, we found that the level of cytoplasmic EBP50 increases progressively from low- through high-grade adenomas to carcinoma lesions, and tracks with the increased malignant potential of colorectal tumor cells. Given that apical localization of EBP50 is required to maintain epithelial integrity and cytoplasmic EBP50 expression increases cell proliferation [5,7], we suggest that perturbation of EBP50 localization is critical for initiating EMT during progression from adenoma to carcinoma. We found that the Cyt-EBP50 score was significantly higher in normal colorectal mucosa as compared to low-grade tubular adenoma. This may be because the translocation of EBP50 is perturbed during malignant transformation, possibly due to changes in the signaling cascade, EBP50 post-translational modifications, and interacting partners [24]. In contrast to our results, others have found that the transition to predominantly cytoplasmic EBP50 localization occurs later, as tumors progress from adenomas to carcinomas [7]. One possible reason for these discordant results is that different methods were used for the detection of EBP50 (immunohistochemistry versus immunofluorescence).

Although Cyt-EBP50 immunophenotype is a known predictor of poor prognosis for CRC patients [7,25,26], we were unable to link Cyt-EBP50 to any specific clinicopathological factor(s) in CRC. In contrast, the EBP50 score was significantly lower in outer lesions, particularly in BD compared to non-BD areas. Given our observations, we suggest that reduced EBP50 expression (rather than overexpression or changes in EBP50 subcellular distribution) may play an important role in CRC progression.

In addition to the clinical CRC data, we found that EBP50-KO induced an EMT-like phenotype that was accompanied by decreased proliferation, enhanced migration capability, increased Slug expression, and stabilized nuclear β-catenin. These findings are consistent with the negative correlation between EBP50 and nuclear β-catenin scores, and the positive correlation between EBP50 and Ki-67 scores. In addition, expression of Slug (a known effector of EMT-related phenotypes) was positively correlated with nuclear β-catenin status in BD areas of outer tumor regions; this is consistent with nuclear β-catenin-dependent transactivation of the *Slug* gene. Given that BD is as a well-established independent prognostic factor in CRC [27], we suggest that EBP50 depletion contributes to aggressive CRC behavior activating the nuclear β-catenin/Slug axis, leading to EMT and BD formation. This hypothesis is supported by previous findings that EBP50 depletion disrupts epithelial morphogenesis, leading to loss of apico-basal polarity and induction of EMT in CRC cells [25,28]. In addition, migratory cells have a lower proliferation rate in comparison with cells in the tumor core, indicating that EMT engenders an inverse correlation between proliferation and mobility [29,30,31].

EMT promotes stem cell properties and confers cells with CSC-like features [32]. Unexpectedly, however, EBP50-KO cells had attenuated CSC properties, despite having EMT-like features. Given that E-cadherin levels were unchanged in EBP50-KO cells, we suggest that their transition to a full EMT state is incomplete EMT, which would explain the lack of CSC properties. Further studies to address this point are clearly warranted. Finally, we found no association between EBP50 mRNA expression and prognosis in CRC patients. Since EBP50 depletion in outer tumor regions was significantly associated with several unfavorable prognostic factors in our CRC series, we surmise that a region-specific analysis of EBP50 expression may be required if this protein is to be used as a prognostic marker in CRC.

## 5. Conclusions

Together, our results suggest a novel role of EBP50 and nuclear β-catenin in CRC progression (Figure 6D). In outer tumor lesions, EBP50 depletion releases β-catenin from the plasma membrane, allowing β-catenin to accumulate and translocate to the nucleus, where it transactivates the *Slug* gene to promote EMT. This in turn triggers tumor budding and contributes to the progression of CRC to a more aggressive phase. Additional in vivo model assays including experiments with CRC cell xenografts in the context of EBP50 overexpression, and knockout will be required to further validate the findings of this study. We predict that knockout of EBP50 should exhibit aggressive tumor behaviors in such in vivo models.

## Figures and Tables

**Figure 1 cancers-16-00183-f001:**
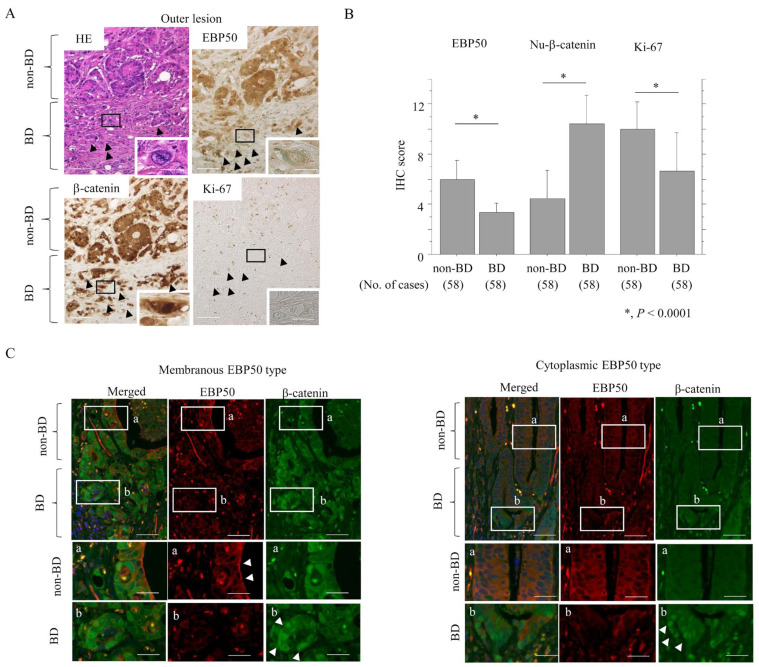
EBP50 depletion in BD areas of outer lesions: (**A**) Staining with HE and IHC for the indicated proteins in tumor budding (BD) and non-BD areas in outer lesions. Note the decreased immunoreactivity for EBP50 and Ki-67 (indicated by arrowheads) in BD areas, in contrast to the increased nuclear β-catenin accumulation (indicated by arrowheads). The closed boxes in upper panels are magnified in insets. Original magnification, ×100 and ×400 (insets). Scale bar = 50 μm and 10 μm. (**B**) IHC scores for the indicated proteins between BD and non-BD areas in outer lesions. The scores shown are means ± SDs. Statistical analyses were carried out using the Mann–Whitney U-test. No., number. (**C**) Double immunofluorescence analysis for the indicated proteins in BD and non-BD areas in outer lesions. Note the membranous staining of EBP50 (indicated by arrowheads: non-BD area in membranous EBP50 type) and the nuclear β-catenin accumulation (indicated by arrowheads: BD areas in membranous and cytoplasmic EBP50 type). Closed boxes (a,b) in the upper panels are magnified in the middle (a) and lower panels (b). Original magnification, ×100 and ×400 (insets). Scale bar = 50 μm and 10 μm.

**Figure 2 cancers-16-00183-f002:**
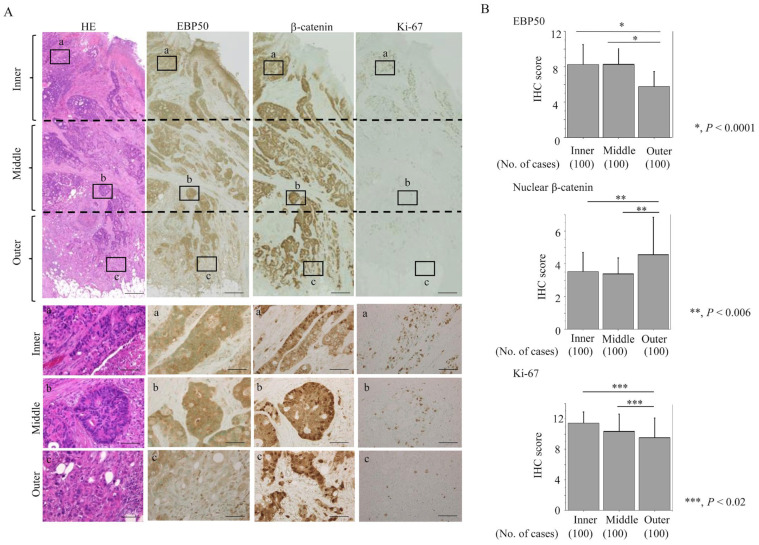
EBP50 depletion in outer lesions of CRC: (**A**) Staining with HE and IHC for the indicated proteins. Note the decreased immunoreactivity for EBP50 and Ki-67 in outer lesions as compared to inner and middle areas, in contrast to the increased nuclear β-catenin accumulation in the former. Closed boxes (a,b,c) in the upper panels are magnified in the lower panels (a,b,c). Original magnification, ×2 and ×200 (lower three panels). Scale bar = 10 μm (upper three panels) and 50 μm (lower three panels). (**B**) IHC scores for the indicated proteins in inner, middle, and outer lesions. The scores shown are means ± SDs. Statistical analyses were carried out using the Mann–Whitney U-test. No., number.

**Figure 3 cancers-16-00183-f003:**
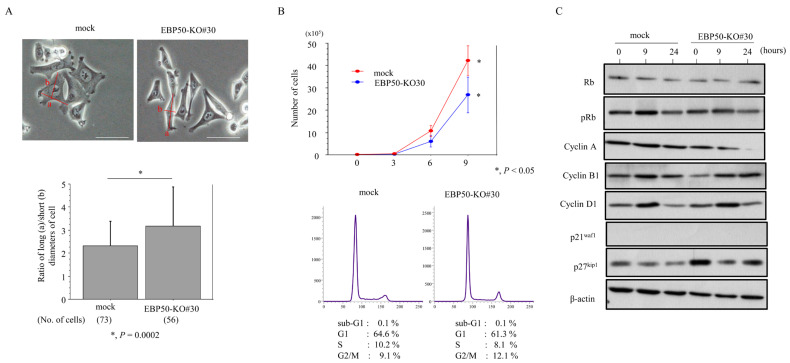
Changes in morphology and proliferation following EBP50 knockout in CRC cells: (**A**) Upper: phase contrast images of EBP50-KO#30 cells, revealing switch towards a fibroblastic morphology. Scale bar = 30 μm. Lower: ratios of long (‘a’ in upper panels) and short (‘b’ in upper panels) diameters in cells. The values shown are means ± SDs. Statistical analyses were carried out using the Mann–Whitney U-test. No., number. (**B**) Upper: EBP50-KO#30 and mock cells were seeded at low density. Cell numbers are presented as means ± SDs. P0, P3, P6, and P9 are 0, 3, 6, and 9 days after seeding, respectively. The experiments were performed in triplicate. Statistical analyses were carried out using the Mann–Whitney U-test. Lower: flow cytometry analysis of EBP50-KO#30 and mock cells 3 days after seeding (P3). (**C**) Western blot analysis for the indicated proteins in total lysates from EBP50-KO#30 and mock cells following re-stimulation of serum-starved (24 h) cells with 10% serum for the indicated times. The uncropped blots are shown in Appendix A.

**Figure 4 cancers-16-00183-f004:**
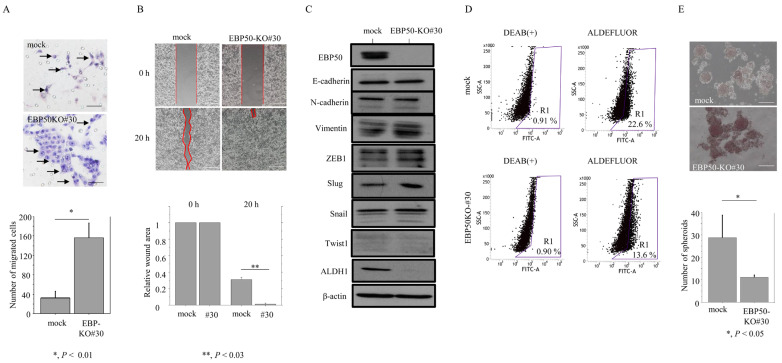
Changes in migration but not cancer stem cell properties in EBP50-KO cells: (**A**) Upper: EBP50-KO and mock cells were seeded in 24-well Transwell plates and incubated for 24 h in medium without serum. Cells (indicated by arrows) were stained with HE and counted using a light microscope. Lower: numbers of migrated cells are presented as means ± SDs. The experiments were performed in triplicate. Statistical analyses were carried out using the Mann–Whitney U-test. Scale bar = 30 μm. (**B**) Upper: wound-healing assay with EBP50-KO#30 and mock cells. A scratch ‘wound’ was introduced to the middle of wells containing cells grown to confluency, and phase contrast images were taken after 20 h. Scale bar = 50 μm. Lower: the values of wound areas at 0 h were set as 1. The fold wound areas are presented as means ± SDs. The experiments were performed in triplicate. Statistical analyses were carried out using the Mann–Whitney U-test. #30, EBP50-KO#30. (**C**) Western blot analysis for the indicated proteins in total lysates from EBP50-KO#30 and mock cells. The experiments were performed in duplicate. The uncropped blots are shown in Appendix A. (**D**) Aldefluor analysis in EBP50-KO#50 and mock cells. Region R1 includes the ALDH^high^ population with cancer-stem-cell-like features. The experiments were performed in triplicate. (**E**) Upper: phase contrast photographs of spheroids formed by EBP50-KO#30 and mock cells following 2 weeks of growth. Scale bar = 50 μm. Lower: spheroid numbers are presented as means ± SDs. The experiments were performed in triplicate. Statistical analyses were carried out using the Mann–Whitney U-test.

**Figure 5 cancers-16-00183-f005:**
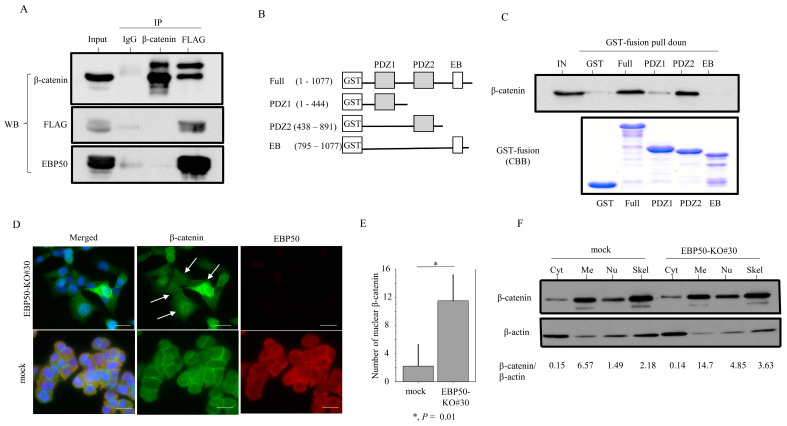
Interaction between EBP50 and β-catenin in CRC cells: (**A**) Western blot (WB) with anti-β-catenin (upper panel), anti-FLAG (middle panel), and anti-EBP50 antibodies (lower panel) after immunoprecipitation (IP) with the indicated antibodies using HCT116 cell lysates. Input represents 5% of the total cell extract. Normal mouse IgG was used as a negative control. (**B**) Schematic representation of the cytosolic PSD-95/*Drosophila* discs large/ZO-1 (PDZ) and EB domains of EBP50. (**C**) Upper: proteins bound to the beads were analyzed followed by Western blot analysis for β-catenin in HCT116 cells. Lower: detection of GST-bound EBP protein by Coomassie Brilliant Blue (CBB). The experiments were performed in duplicate. (**D**) Double immunofluorescence analysis for the indicated proteins. Note the nuclear β-catenin accumulation (indicated by arrows) in EBP50-KO#30 cells (upper panels), in contrast to colocalization of EBP50 and β-catenin at membranous sites in mock cells (lower panels). Scale bar = 20 μm. (**E**) Numbers of cells with nuclear β-catenin accumulation are presented as means ± SDs. The experiments were performed in triplicate. Statistical analyses were carried out using the Mann–Whitney U-test. (**F**) Western blot analysis for the indicated proteins in cytoplasmic (Cyt), membranous (Me), nuclear (Nu), and cytoskeletal fractions (Skel). Ratios of β-catenin relative to β-actin are calculated using ImageJ version 1.41. The uncropped blots are shown in Appendix A.

**Figure 6 cancers-16-00183-f006:**
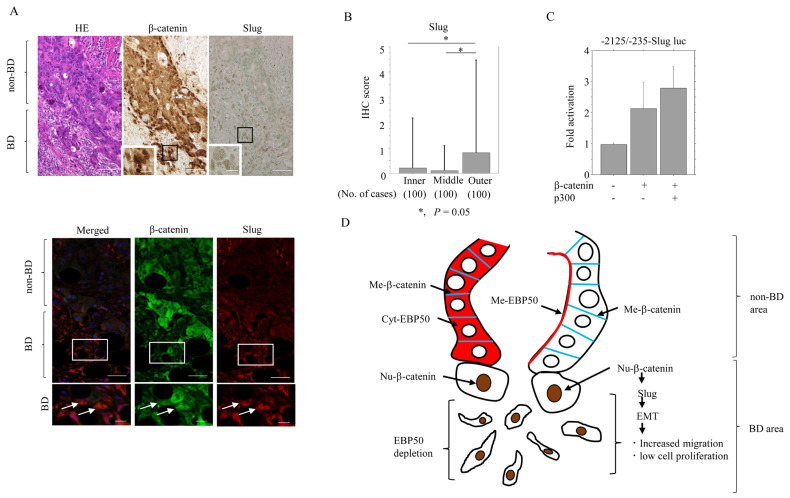
Relationship between Slug and β-catenin expression in BD areas of outer lesions: (**A**) Upper: staining with HE and IHC for the indicated proteins in CRC. Note the increased immunoreactivity for nuclear β-catenin and Slug in BD areas. The closed boxes are magnified in the insets. Original magnification, ×100 and ×400 (insets). Scale bar = 50 μm and 10 μm (insets). Lower: double immunofluorescence analysis for the indicated proteins. Red, cytoplasmic or membranous EBP50 expression; Blue, membranous beta-catenin; Brawn, nuclear beta-catenin. Note the colocalization of nuclear β-catenin and Slug in BD areas (indicated by arrows). The closed boxes are magnified in the insets. Original magnification, ×100 and ×400 (insets). Scale bar = 50 μm and 10 μm (insets). (**B**) IHC scores for Slug in the inner, middle, and outer lesions. The scores shown are means ± SDs. Statistical analyses were carried out using the Mann–Whitney U-test. No., number. (**C**) HCT116 cells were transfected with *Slug* promoter constructs, together with β-catenin and p300. Relative activity was determined based on arbitrary light units of luciferase activity normalized to pRL-TK activity. The activities of the reporter plus the effector relative to that of the reporter plus empty vector are shown as means ± SDs. The experiments were performed in triplicate. (**D**) Schematic representation of the interplay between EBP50 and nuclear (Nu) β-catenin during CRC progression. Me, membranous, Cyt, cytoplasmic, Nu, nuclear, BD, tumor budding, EMT, epithelial–mesenchymal transition.

**Table 1 cancers-16-00183-t001:** Relationship between EBP50 expression and clinicopathological factors in colorectal carcinoma.

		EBP50 Expression				EBP50 Depletion in Outer Lesion		
		Membrane	Cytoplasm			Severe to Mod	None/Mild	
	*n*	*n* (%)	*n* (%)	*p*-Value	*n*	*n* (%)	*n* (%)	*p*-Value
Gender								
Male	58	25 (43.1)	33 (56.9)	0.7	58	35 (60.3)	23 (39.7)	0.5
Female	42	16 (38.1)	26 (61.9)		42	29 (69)	13(31)	
Age								
≤68 years	39	17 (43.6)	22 (56.4)	0.8	39	27 (69.2)	12 (30.8)	0.4
≥69 years	61	24 (39.3)	37 (60.7)		61	37(60.7)	24 (39.3)	
Tumor location								
Right	23	7 (30.4)	16 (69.6)	0.3	23	17 (73.9)	6 (26.1)	0.3
Left	77	34 (44.2)	43 (55.8)		77	47 (61)	30 (39)	
Macroscopic finding								
Depression	94	38 (40.4)	56 (59.6)	0.9	94	64 68.1)	30 (31.9)	0.003
Elevation	6	3 (50)	3 (50)		6	0 (0)	6 (100)	
Hist. differentiation								
tub1	47	24 (51.1)	23 (48.9)	0.1	47	31 (66)	16 (34)	0.1
tub2	53	17 (32.1)	36 (67.9)		53	33 (62.3)	20 (37.7)	
Tumor size								
≤4.2 cm	55	24 (43.6)	31 (56.4)	0.6	55	34 (61.8)	21 (38.2)	0.7
≥4.3 cm	45	17 (37.8)	28 (62.2)		45	30 (66.7)	15 (33.3)	
Depth								
less than SS	76	31 (13.6)	45 (59.2)	0.9	76	44 (57.9)	32 (42.1)	0.04
SE	24	10 (41.7)	14 (58.3)		24	20 (83.3)	4 (16.7)	
Budding								
BD1	41	15 (36.6)	26 (63.4)	0.5	42	17 (40.5)	25 (59.5)	<0.0001
BD2/3	59	26 (44.1)	33 (55.9)		58	47 (81)	11 (19)	
Ly invasion								
Positive	41	10 (24.4)	31 (75.6)	0.009	41	29 (70.7)	12 (29.3)	0.3
Negative	59	31 (52.5)	28 (47.5)		59	35 (59.3)	24 (40.7)	
V invasion								
Positive	74	29 (39.2)	45 (60.8)	0.6	74	50	24	0.3
Negative	26	12 (46.2)	14 (53.8)		26	14	12	
Neural invasion								
Positive	44	14 (31.8)	30 (68.2)	0.1	44	35 (79.5)	9 (20.5)	0.007
Negative	56	27 (48.2)	29 (51.8)		56	29 (51.8)	27 (48.2)	
LN metastasis								
Positive	56	21 (37.5)	35 (62.5)	0.5	56	38 (67.9)	18 (32.1)	0.4
Negative	44	20 (45.5)	24 (54.5)		44	26 (59.1)	18 (40.9)	
Distant metastasis								
Positive	15	3 (20)	12 (80)	0.1	15	12 (80)	3 (20)	0.2
Negative	85	38 (44.7)	47 (55.3)		85	52 (61.2)	33 (38.8)	
Clinical stage								
I/II	41	19 (46.3)	22 (53.7)	0.2	41	23 (56.1)	18 (43.9)	0.2
III/IV	59	22 (37.3)	37 (62.7)		59	41 (69.5)	18 (30.5)	
EBP50 depletion								
severe to moderate	64	23 (35.9)	41 (64.1)	0.1		NE	NE	NE
none/mild	36	18 (50)	18 (50)			NE	NE	

SS, subserous; SE, serous exposure; Ly, lymph; V, venous; LN, lymph node; BD, tumor budding; NE, not examined. Statistical analyses were carried out using the Chi-square test.

**Table 2 cancers-16-00183-t002:** Correlation between EBP50 and related markers in outer lesions of colorectal carcinoma.

	EBP50	Nu-β-Catenin	Ki-67
	*ρ* (*p*)	*ρ* (*p*)	*ρ* (*p*)
Nu-β-catenin	−0.49	*	*
	(<0.0001)		
Ki-67	0.59	−0.33	*
	(<0.0001)	(0.0004)	
Slug	0.33	0.44	0.37
	(0.0005)	(<0.0001)	(<0.0001)

*ρ*, Spearman’s correlation coefficient; *, not examined; Nu, nuclear; Statistical analyses were carried out using the Spearman’s correlation coefficient.

## Data Availability

The data sets generated during and/or analyzed during the current study are available from the corresponding author on reasonable request.

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
