# Peer review of "EBP50 Depletion and Nuclear β-Catenin Accumulation Engender Aggressive Behavior of Colorectal Carcinoma through Induction of Tumor Budding"

_cancers, 2023, doi:10.3390/cancers16010183_

Round 1
Reviewer 1 Report
Comments and Suggestions for Authors
Itou et al found that EBP50 expression was lower in the outer area of CRC tumor lesions, especially in the budding region. Depletion of EBP50 was associated with nuclear accumulation of β-catenin, promotion of EMT and enhancement of cell migration. Finally, the authors proposed a model in which EBP50 depletion releases β-catenin into nucleus, activates EMT and triggers tumor budding. The manuscript is written smoothly and of great interests to readers in the field.
Major points:
1) For EBP50 KO experiments, have the authors tested in the other two cell lines mentioned in the method section? It would be more convincing by providing evidence in additional cell lines.
2) Figure 4 showed some of these markers were altered by EBP50 depletion in HCT116 cells. Did the authors performed IHC scores of EMT or stem cell markers and compared BD and no-BD, outer and inner, on clinical samples?
3) For all bar graphs please indicate the samples size in each group.
4) Please provide full images of western blot, not cropped, in the supplementary materials.
5) The authors may consider including results section 3.5 “EBP50 expression is not a prognostic indicator in CRC” into 3.1 “EBP50 expression during adenoma-carcinoma progression” as it’s more relevant.
Minor
1) The location of Figure 1 and Figure 2 need to be swapped.
2) Please proofread the manuscript once more. For example, “marged” in Figures 2, 6 should be “merged”.
3) Please specify catalog numbers for antibodies used in methods section 2.2.
Comments on the Quality of English Language
Please proofread the manuscript.
Author Response
Major points:
Suggestion 1) For EBP50 KO experiments, have the authors tested in the other two cell lines mentioned in the method section? It would be more convincing by providing evidence in additional cell lines.
Answer:
As answered to the above Academic Editor’s comment, we were unable to establish EBP50 KO cells using DLD1 and OUMS22 cells.
Suggestion 2) Figure 4 showed some of these markers were altered by EBP50 depletion in HCT116 cells. Did the authors performed IHC scores of EMT or stem cell markers and compared BD and no-BD, outer and inner, on clinical samples?
Answer:
Based on the findings that EBP50-KO#30 cells increased expression of the EMT-related molecules, Slug, but not Snail, ZEB1, and Twist1, as described in the Results (page 13, 2nd paragraph), we investigated an association between Slug and nuclear ß-catenin expression in clinical samples and cell lines of CRC. The other EMT/CSC-related markers were not applied, because there were not differences in the expression of these markers between the KO and mock cells, with the exception of ALDH1 status.
Suggestion 3) For all bar graphs please indicate the samples size in each group.
Answer:
Number of samples investigated are newly provided in most bar graphic figures including Figures 1B, 2B, 3A, and 6B.
Suggestion 4) Please provide full images of western blot, not cropped, in the supplementary materials.
Answer:
To detect expression of several proteins by western blot assay, the membranes were cut according to the predictive molecular weights of the proteins. The Supplementary materials show the reconstruction of separated membranes.
Suggestion 5) The authors may consider including results section 3.5 “EBP50 expression is not a prognostic indicator in CRC” into 3.1 “EBP50 expression during adenoma-carcinoma progression” as it’s more relevant.
Answer:
The description of the results section 3.5 is now transferred to the results section 3.1 as follows:
Page 11, last paragraph
3.1 EBP50 expression during adenoma-carcinoma progression and TCGA data analysis in CRC
EBP50 immunoreactivity was observed in the apical plasma membrane (Me-EBP50) and/or cytoplasmic compartment (Cyt-EBP50) of epithelial cells during adenoma to carcinoma progression in CRC (Supplementary Figure S2A). Me-EBP50 scores did not correlate with the grade of adenoma. In contrast, there was a stepwise increase in Cyt-EBP50 scores from low- through high-grade adenomas to carcinoma in adenoma (non-invasive CRC) lesions. Cyt-EBP50 score was also significantly higher in normal as compared to low-grade adenoma lesions (Supplementary Figure S2B). Although there was a heterogeneous distribution of Me- and Cyt-EBP50 positivity within CRC tissues (Supplementary Figure S2C), Me- and Cyt-EBP50 immunopositivity predominated in 41% and 59% of the 100 CRC cases, respectively. With the exception of tumor invasion into lymph vessels, neither membrane nor cytoplasmic localization of EBP50 correlated with the clinicopathological samples we examined (Table 1). In addition, we used TCGA data to evaluate the prognostic significance of EBP50 expression in 588 CRC cases. Following separation of cases into EBP50-high and -low group, Kaplan-Meier curves analysis showed that EBP50 expression had no prognostic significance for OS or PFS (Supplementary Figure S3).
Minor
Suggestion 1) The location of Figure 1 and Figure 2 need to be swapped.
Answer:
The locations of Figures 1 and 2 are now exchanged. The Results section is also revised as follows:
Page 12, line 4
3.2 Reduced EBP50 expression in outer lesions of CRC is associated with aggressive tumor behavior
Decreased EBP50 immunostaining was frequently observed in BD areas of outer lesions (Figure 1A), where it was accompanied by strong nuclear ß-catenin and a lack of Ki-67 immunostaining. This observation was reflected by IHC scores (Figure 1B, C), confirming that EBP score was positively and negatively correlated with Ki-67 and nuclear b-catenin scores, respectively (Table 2).
We next examined the correlations between EBP50, ß-catenin, and Ki-67 staining and determined whether there was region-specific expression of these markers. EBP50 (including Me- and/or Cyt-EBP50 subtypes) and Ki-67 scores were significantly lower in the outer than the inner and middle areas of lesions, in contrast to the significantly higher nuclear ß-catenin scores in the outer area (Supplementary Figure S1, Figure 2). In the outer lesions, 64% of cases had high (severe/moderate) EBP50 depletion and 36% had low (mild/none) EBP50 depletion. Compared to the low EBP50 depletion group, the high EBP50 depletion group was significantly associated with depressive tumor growth, deep invasions, high BD scores, and the presence of neural invasions. The level of EBP50 expression between Me- and Cyt-EBP50 categories was similar (Table 1).
Suggestion 2) Please proofread the manuscript once more. For example, “marged” in Figures 2, 6 should be “merged”.
Answer:
“Marged” in Figures 2 and 6 were changed to “Merged”. In addition, we now proofread the manuscript again.
Suggestion 3) Please specify catalog numbers for antibodies used in methods section 2.2.
Answer:
The catalog numbers for antibodies used in this study are newly provided in the Methods section 2.2, as follows:
Page 6, first paragraph
2.2 Antibodies
Anti-phospho (p) Rb at Ser807/811 (catalog number #9308) and anti-Slug antibodies (#9585) were purchased from Cell Signaling Technology (Danvers, MA, USA). Anti-EBP50 (ab109430), Anti-cyclin A2 (#ab181591), anti-Snail (#ab180714), and anti-Twist1 (#ab50887) antibodies were obtained from Abcam (Cambridge, MA, USA). Anti-E-cadherin, anti-Ki-67 (#M7240), anti-p21waf1 (#M7202), and anti-cyclin D1 antibodies (#M3635) were from Dako (Copenhagen, Denmark). Anti-p27kip1 (#610242), anti-Rb (#554136), anti-aldehyde dehydrogenase 1 (ALDH1) (#611194), anti-β-catenin (#610154), and anti-N-cadherin antibodies (#610920) were from BD Biosciences (San Jose, CA, USA). Anti-FLAG M2 (#F3165), anti-ZEB1 (#HPA027524), and anti-b-actin antibodies (#HPA025958) were from Sigma-Aldrich Chemicals (St. Louis, MO, USA). Anti-vimentin (sc-6260) and anti-cyclin B1 antibodies (#sc-752) were from Santa Cruz Biotech (Santa Cruz, CA, USA). Anti-E-cadherin antibody (#M106, Takara Bio Inc, Siga, Japan) was purchased from Cell Signaling Technology (Danvers, MA, USA).
Reviewer 2 Report
Comments and Suggestions for Authors
Thanks for inviting me to evaluate the article titled ‘EBP50 depletion and nuclear beta-catenin accumulation engender aggressive behavior of colorectal carcinoma through induction of tumor budding. In this article, the authors found that EBP50 depletion is due to 19 nuclear beta-catenin accumulation, allowing transactivation of Slug gene to promote EMT, which in turn contributes to the progression of CRC to a more aggressive phase through induction of tumor budding. At the beginning, the simple summary and the abstract are almost the same, maybe the authors may figure out the emphasis of each part and make revisions. And I still have some concerns. Why the location of EBP50 may have effect on the phenotype of CRC? Why knockout of EBP50 decreased proliferation while accelerated migration? Can we do some animal study to better understand the function of EBP50 in CRC? In addition, the authors mentioned that ‘Similar findings were also reported in other human malignancies including ovarian, biliary, and breast carcinomas’, could you please explain this more detailed?
Comments on the Quality of English Languagethe english of this paper is easy to read and of high quality.
Author Response
Suggestion 1) At the beginning, the simple summary and the abstract are almost the same, maybe the authors may figure out the emphasis of each part and make revisions.
Answer:
The simple summary is now revised as follows:
Simple summary
Here, we examined the functional role of Ezin-radixin-moesin-binding phosphoprotein 50 (EBP50) during colorectal carcinoma (CRC) progression. EBP50 depletion was frequently found in the outer areas of tumor lesions and was significantly associated with several unfavorable prognostic factors including depressive tumor growth, deep invasions, high tumor budding (BD), and the presence of neural invasion in CRC cases. EBP50-knockout CRC cells disrupted the interaction between EBP50 and ß-catenin at the plasma membrane, leading to epithelial-mesenchymal transition (EMT)-like features, decreased proliferation, accelerated migration capability, and stabilized nuclear ß-catenin. In addition, Slug expression was positively correlated with nuclear ß-catenin status in tumor outer lesions including BD areas, consistent with ß-catenin-driven transactivation of the Slug promoter. These findings suggest that EBP50 depletion is due to nuclear ß-catenin accumulation, allowing transactivation of Slug gene to promote EMT. This in turn contributes to the progression of CRC to a more aggressive phase through induction of tumor budding.
Suggestion 2) Why the location of EBP50 may have effect on the phenotype of CRC?
Answer:
Given that apical localization of EBP50 is required to maintain epithelial integrity and cytoplasmic EBP50 expression increases cell proliferation [5,7], we consider that perturbation of EBP50 localization is critical for initiating EMT during progression from adenoma to carcinoma as described in the Discussion (page 15, 2nd paragraph). In addition, EBP50 depletion also contributes to aggressive CRC behavior through activating the ß-catenin/Slug axis, leading to EMT and BD formation as described in the Discussion (page 16, 2nd paragraph).
Suggestion 3) Why knockout of EBP50 decreased proliferation while accelerated migration?
Answer:
This may be supported by evidence that migratory cells have a lower proliferation rate in comparison with cells in the tumor core, indicating that EMT engenders an inverse correlation between proliferation and mobility [32-34] as described in the Discussion (page 16, line 11 from the bottom).
Suggestion 4) Can we do some animal study to better understand the function of EBP50 in CRC?
Answer:
We agree with the reviewer’s comment that validation using in vivo models would further strengthen the association between EBP50 and phenotypic characteristics in CRC cells. However, we are unable to carry out in vivo assays due to the deadline (within 10 days) for resubmission of the revised manuscript, which is requested by Section Managing Editor. Therefore, we have added a sentence regarding the importance of in vivo models to the Discussion as follows:
Page 17, line 10
Additional in vivo model assays including experiments with CRC cell xenografts in the context of EBP50 overexpression and knockout will be required to further validate the findings of this study. We predict that knockout of EBP50 should exhibit aggressive tumor behaviors in such in vivo models.
Suggestion 5) In addition, the authors mentioned that ‘Similar findings were also reported in other human malignancies including ovarian, biliary, and breast carcinomas’, could you please explain this more detailed?
Answer:
Because the description is an inappropriate explanation for the association between EBP50, nuclear ß-catenin, and Ki-67 scores, we now delete the sentence, as well as the references.
Round 2
Reviewer 2 Report
Comments and Suggestions for Authors
Thanks for inviting me to evaluate the article titled ‘EBP50 depletion and nuclear beta-catenin accumulation engender aggressive behavior of colorectal carcinoma through induction of tumor budding. In this article, the authors found that EBP50 depletion is due to 19 nuclear beta-catenin accumulation, allowing transactivation of Slug gene to promote EMT, which in turn contributes to the progression of CRC to a more aggressive phase through induction of tumor budding. The authors have answered all the questions I have raised. Thus, this paper could be accepted in its present form.
Comments on the Quality of English LanguageYou do not need to revise the quality of the English language.